# Barking Up the Wrong Tree—Motor–Sensory Elements as Prodrome in Autism

**DOI:** 10.3390/biomedicines12061235

**Published:** 2024-06-02

**Authors:** Meir Lotan

**Affiliations:** Physical Therapy Department, Ariel University, Ariel 40700, Israel; meirlo@ariel.ac.il

**Keywords:** autism, ASD, motor elements, early diagnosis, early intervention

## Abstract

Autism spectrum disorder (ASD) has been intensely investigated since the term was first used over 80 years ago. The prevalence of ASD is constantly rising, and, currently, 1:36 children are diagnosed with this disorder. Despite the intense interest in ASD, the origins of this disorder remain obscure. This article explores motor issues and proprioceptive interoception difficulties as the prodrome of ASD. The importance of early intervention in the prognosis of ASD is common knowledge. Yet, since the communicational and social behaviors typical of ASD are observable only after the age of 18 months, diagnosis and early intervention are delayed. Therefore, the quest into the involvement of sensory–motor difficulties as a source of ASD traits, or at least as a potential early indicator, is warranted, with the intention of enabling early diagnosis and early intervention. This article examines the justification for this new avenue of early diagnosis and intervention and may open up a completely different way of viewing ASD. This new point of view may suggest an original path of assessment and intervention in infancy with this group of clients, possibly leading to improved prognosis for children and their families.

## 1. Autism—Introduction

Autism Spectrum Disorder (ASD) is defined by the Diagnostic and Statistical Manual of Mental Disorders, fifth edition, as having impairments in social interactions, communication, and speech development, and repetitive behaviors with restricted interests [1]. In the last three decades, the number of reported ASD cases has increased from 1 in 2000–2500 [2] to approximately 1 in 36 of children diagnosed in the USA, making it the developmental disorder with the highest incidence rate [3]. ASD is much more common in males than in females, with an approximate ratio of 3–4:1, although some researchers believe that there may be underdiagnosis concerning females [4].

Despite intensive ongoing ASD research, the cause for this disorder remains enigmatic, with numerous potential causes and conflicting variables being linked to it. Suggested causes include genetic components, environmental causes, parental age, maternal nutrition during pregnancy, and bio-neural factors, such as brain structure and development [1].

In line with the accepted definition of ASD today [1], the main symptoms of ASD are difficulties in social communication or social interaction, such as social–emotional reciprocity, nonverbal communicative behavior, or relationships in general [3]. Repetitive or restrictive behavior or interests may also be present. To meet the definition, these behaviors must appear early in life and create a social impairment at some level. Therefore, current testing and diagnosis of ASD is based on the clinical manifestation of the abovementioned social, behavioral, and communicative difficulties and challenges in specific skills.

Early treatment is known to improve the prognosis of autistic children [5]; therefore, early diagnosis is crucial. Nonetheless, the diagnosis of autism is currently made between the ages of 3 and 10 years worldwide [6], due to the reliance on social and communicative elements that are fully developed only within the second year of life. Despite the fact that motor symptoms have recently been the focus of intense research [7,8,9], they are not yet part of the official symptoms listed in DSM5 and are not used as diagnosis criteria [1].

A new and emerging notion raises the possibility that a combination of interoception misinterpretation, as well as motor issues, stand at the core of ASD. This opinion article focuses on the sensory–motor issues of ASD individuals and suggests a new diagnostic direction that considers sensory–motor elements as the core of ASD. To date, motor elements have been considered a minor element affecting some individuals with ASD, without any major influence on the functional abilities or the source of ASD. The next section explores the potential role of motor issues as a core, or at least as a root, cause of ASD.

## 2. Sensory–Motor Misinterpretation as a Cause of ASD

Kanner reported in an original article that these individuals present motor clumsiness and an odd gait [10]. This observation was later reiterated by Asperger (1944) [11], who suggested that the difficulty in performing activities of daily living (ADL) by individuals with ASD may be due to their motor difficulties and not merely their autistic traits. Eighty years later, motor clumsiness and odd gait are reiterated as secondary elements associated with ASD in DSM-5 [1]. Despite suggestions by distinguished authors in the field, such as Teitelbaum et al. [12], who stated that “In infancy, the movement disorders presented in autism are clearest, not yet masked by other mechanisms that have developed to compensate for them”, the motor elements of ASD are mostly ignored by many researchers and clinicians. To start this academic quest, we first need to determine whether the vast majority of individuals with ASD experience motor difficulties as a core issue.

This article explores the basis for this novel opinion that sensory–motor issues are the cause of ASD by addressing the following questions:How common are motor issues in the population of ASD?At what age can motor irregularities be observed in infants later diagnosed with ASD?How do scholars explain motor difficulties as the basis of the later appearance of social and communicative difficulties?Can motor problems be interpreted as social\communicational\behavioral issues by an external observer?How do individuals with ASD describe their motor experiences?What are the effects of motor-based interventions on the core ASD elements?

### 2.1. How Common Are Motor Issues in the Population of ASD?

Over the past two decades, motor components have increasingly become a focus of attention in ASD research [7,8,9]. It is now known that motor and coordination deficiencies are present in about 87% of individuals with ASD [13], and many have difficulties in their ability to execute coordinated motor actions, resulting in slow, clumsy, or inaccurate motor performance and motor-learning difficulties. Unfortunately, most children with ASD will never outgrow their motor difficulties [13]. Despite this high percentage and suggestions that motor elements are inherently related to core ASD traits [14], the diagnostic criteria for ASD in DSM5 do not include any motor components [1,15].

Moreover, most research on this issue suggests that motor challenges exacerbate with age, and therefore by the age of 9–12 years, children diagnosed with ASD perform motor tasks at a level of children half their age [16] and present an array of motor abnormalities such as difficulty coordinating rotational trunk movements or controlling their limbs’ movements. Body segments appear to be acting independently, and this predicament is clinically observed as a stiff gate, over-fluidity of walking, and reduced smoothness of movement and posture [17]. At this point, we have established that motor difficulties are a major issue for children diagnosed with ASD, which does not vanish with age; yet, if we are to use such elements as a key to early diagnosis of ASD, these elements must be presented from an early age. Therefore, we must determine the age at which age motor abnormalities can be observed in this population.

### 2.2. At What Age Can Motor Irregularities Be Observed in Autistic Infants?

Multiple studies have been conducted using video analysis of infants later diagnosed with ASD. Esposito et al. [8] analyzed home videos of children aged 4–5 months in a supine position who were later diagnosed with ASD. These videos were compared to infancy videos of children presenting typical development. The researchers’ findings suggest that infants who were later diagnosed with ASD presented greater asymmetry of posture and movement compared to typically developing babies. Very similar results were obtained in a study that found that less rhythmic and weaker bodily movement patterns at 4 months of age were typical of children later diagnosed with ASD, in comparison to typically developing infants [7]. Another research project also used video analysis of infants as young as 0–4 months of age and evaluated the appearance of general movements (GM) in both groups [9]. These researchers found that 70% of the infants who were later diagnosed with ASD presented with bradykinesia and abnormal GM, in contrast to the age-appropriate GM of typically developing infants. Since motor development starts from the gestational period and does not rely on social or linguistic development [8,18], the existence of such motor abnormalities at such a young age is an alarming phenomenon. If such peculiarities exist, independent of any connection to social and communicative abilities, this suggests that deficits in motor elements are present in infants with ASD from birth. This abnormality raises the possibility that motor issues are a key element in ASD. This possibility highlights the importance of movement analysis for infants in the first months of life [14], in order to detect early motor indications that will enable early diagnosis and early intervention for infants later diagnosed with ASD [19]. These results are also supported by findings that motor development at age 6 months was correlated with ASD status at age 24–36 months, suggesting the possibility that ASD is associated with awkward and abnormal early motor performance [20].

The next question to be asked is, what is the connection between early motor deficiencies and the later-evolving communicational and social difficulties, which are, at present, the hallmark of ASD?

### 2.3. How Can Sensory–Motor Challenges in Infancy Affect Communication and Social Challenges Later Observed as Clinical Characteristics of Individuals with ASD?

This section presents the conclusions of researchers who investigated early mother–infant relationships and interactions. Since movement is a physical expression of one’s psychological being, to the external observer, the minute nuances of one’s body movements represent one’s intentions and emotions. Our body movements also represent our mutual experiences in a common social environment [21]. Therefore, an autistic infant, later to be diagnosed with ASD, will have a hard time understanding the messages conveyed by their parents as well as by their own body. Those difficulties are later manifested by the babies’ inability to control their bodily responses to incoming stimuli. The outcome will develop into a bilateral misunderstanding of physical messages (facial gestures, body movements): those sent to them by others, as well as difficulty in externalizing their own self-messages. Greenspan attests to the fact that the capacity for empathy, emotional and mental ability, abstract thinking, social problem solving, and functional language (which present the core elements of ASD) are all key elements in achieving normative interrelationships. Yet, they are **all** dependent on an infant’s ability **to connect their inner emotions with motor planning** when involved in those initial mother–infant affective interactions [22]. Initial communication between infants and their parents, termed “communicative musicality”, is dependent upon the **exact coordination of rhythms and timing** of two **motor systems.** Effective communicative musicality develops when two systems are focused and oriented toward each other and create a **precise joint interaction** [23]. When this delicate balance is upset, the frequency and quality of these interactions diminish. An infant’s ability to perform a specific motor action in response to an incoming sensory stimulus (termed “reafference”), sets the stage for an exchange of social information between two individuals. This basic ability (which will later develop into conventional communication and social interaction) is forming and developing with practice, towards the other. Such early vocal and physical messages and interactions involving a neuro-typical infant who has no sensory–motor challenges, will later become the corner stone of communicational and an interactional social dyad [24]. Therefore, difficulty in performing these initial communicative musicality encounters (due to the abovementioned sensory–motor difficulties) will subsequently evolve into communication challenges in children with ASD. These difficulties in engaging in such initial motor bilateral tuning, from birth, will have **cascading developmental effects in all areas** at later stages in development [14].

The author of the current article believes that this initial lack of sensory (proprioceptive)–motor ability to activate one’s body in a coordinated manner will gradually diminish (“masked by other behaviors” [12]). This occult difficulty and the compensations for this initial difficulties resurface at age 18 months and onward. At this time age-appropriate social and communicational tasks are expected to be interpreted as communicational/lingual difficulties and are termed at that point as ASD.

The diagnosis of children with ASD due to their difficulties in communicating and interacting as such is based on the assumption that emotional/communicative expression is distinct from motor expression. To refute this assumption, we need to examine whether motor challenges can be misinterpreted by external observers as communication and social issues.

### 2.4. Can Motor Problems Be Interpreted as Social/Communication/Behavioral Issues by an External Observer?

Might internal cues that are viewed by external observers and explained within a social, behavioral, or communicative context have a different cause or explanation? This section suggests that observed behaviors (under the titles: emotional, behavioral, social, or communication abnormalities) might be related to sensory–motor challenges presented from infancy by individuals diagnosed with ASD.

We tested this notion using the Modified Rogers Scale. Rogers published an observational checklist [25], later known as the Modified Rogers Scale [26], which was validated by Lund at al. [27]. Below, (see Table 1) we present a section of the Modified Rogers Scale that refers to abnormalities often viewed in children diagnosed with ASD, together with a potential sensory–motor explanation for each abnormality.

The findings so far confirm the existence of motor issues in infants later diagnosed with ASD, from birth, and suggest the important role that intact body–sensory interactions play in initiating the development of communication and socialization. If these findings are correct, then individuals diagnosed with ASD have reported these issues. Therefore, we need to see what individuals diagnosed with ASD are telling us about their own body sensations.

### 2.5. How Do Individuals Diagnosed with ASD Describe Their Motor Experiences?

The next question to be answered pertains to the client’s perspective. This section presents the direct impressions of individuals diagnosed with ASD. In his book “Ido in Autismland”, Kedar addresses his experience of a lack of reliable sense of his body, and discusses the disconnection between his mind and thoughts, which manifests in his body’s failure to respond to motor commands that his brain signals [39]. A similar testimony comes from Tito Mukhopadhyay [40], who stated, “I remember during childhood, I would command my lips to speak, but they would not move”. This phenomenon of “lack of lips moving”, which is currently diagnosed as an impairment in communication and speech development, suggests a clear sensory–motor foundation for this so-called “behavioral/communicative” component of an ASD diagnosis. Mukhopadhyay also attests to the fact that he used to bang his head on objects since he did not know where it was located. Head banging is typically treated through behavioral modification methods; yet, as the difficulty comes from the misinterpretation of sensory information, a different, more effective, clinical approach is needed [40]. In another testimony, Naoki Higashida claims that he does not understand how to move his limbs as they “feel like the rubbery tail of a mermaid” [41]. Similar sensations are reported by others [42,43]. This approach of examining behavioral symptoms from the client’s perspective and through the client’s experience is interesting, as it reveals a clear pattern of sensory–motor issues as the basis of ASD behaviors. Many individuals diagnosed with ASD also suggest solutions to their difficulties. Kedar, for example, says that he believes that his prognosis would have significantly improved if he had been trained, as a child, in agility, fitness, muscle strengthening, and other exercises that would “connect his brain and his body” [39].

It is known that motor skills, especially imitation, are important in the development of play, interaction with others, communication, and language [44]. Sensory–motor deficits increase with age, and the gap in motor abilities between the child with ASD and their typically developing peers increases over time [45]. If the hypothesis of the current article is correct, motoric interventions with individuals diagnosed with ASD should improve the core issues of ASD. Therefore, the next section of this article assesses this hypothesis by asking whether motoric interventions can improve core ASD difficulties such as behavioral, communicative, language, and sensory issues, and stereotypical behaviors, as well as comorbidities typical for this population, including sensory issues, reduced quality of life, and even ASD severity.

### 2.6. Can Motor-Type Interventions Improve Core Elements of the Autism Diagnosis?

Physical activity has been associated with an array of beneficial outcomes within the general population [46]; however, do individuals and children diagnosed with ASD also benefit from these favorable effects? Many motor-based intervention programs have been developed and implemented with individuals with ASD over the years, including hydrotherapy [47], hippotherapy [48], aerobic exercises [49], karate kata techniques [50], swimming [51], treadmill training [52], outdoor adventure programs [53], trampoline training [54], table tennis [55], and roller skating [56]. Some of these interventions have indicated body-related improvements, including improved balance and flexibility [51], reduction in Body Mass Index (BMI) [52], improved motor proficiency, and lower limb strength [54]. Other general improvements related to physical activity are enhanced wellbeing, ability to interact, and emotional state [57].

Yet, the existing literature also suggests that these motor-based interventions were also associated with positive changes in the core elements of ASD. For example, motor-type interventions were found to **reduce self-stimulation behaviors** [53,56], **stereotypical behaviors** of children with ASD [49,51,53], as well as improved **social behavior**s [47,48,53,55,58], **communication skills** [59], and **sensory regulation** [47,57]. A tendency toward a reduction in **ASD severity** due to outdoor activities was also reported [53]. Positive effects of physical interventions were also reported in co-morbidities typical of ASD, such as a reduction in **challenging behaviors** [49] and **distress behaviors** [57] and an improvement in **academic engagement** [60]. The accumulated evidence thus suggests that various types of physical exercise improve all core elements of ASD, which supports the hypothesis that ASD is actually a motor-based disorder.

## 3. Future Steps to Be Implemented

To implement the ideas raised within the current article, future steps can be taken:More rigorous research should be performed to strengthen (or refute) the current direction, suggesting a significant causable connection between sensory–motor elements and core ASD characteristics.Collect an array of signs and behaviors (with emphasis on motor elements, performed by physical therapists) that can be used to assess early signs of ASD through direct observations of videos of babies later diagnosed with ASD.Collect an array of signs and behaviors (with emphasis on motor elements, performed by physical therapists) that can be used to assess early signs of ASD through focus groups of parents of children diagnosed with ASD.Based on direct observations and focus groups, early identification charts (observation scales for therapists and a parental questionnaire) for ASD should be constructed and applied within baby-follow-up stations, at ages 2–5 months.The observed babies should be followed for 8 years, and the scores and results of the ones diagnosed with ASD should be compared to typically developing children who underwent the same procedure.The scales\observation charts should be re-evaluated, corrected, and validated according to the new diagnosis.The process proposed in point 4 should be repeated, and the babies suspected of ASD and their families should be followed, treated, and supported by a multi-professional team of experts in the field of ASD.The prognosis of the babies receiving the early intervention program should be compared to children diagnosed with ASD, who did not receive the early intervention treatment.

## 4. Conclusions

While the exact cause of ASD is still enigmatic, several factors have been proven to influence the potential diagnosis and treatment. Research findings suggest that the DSM5 does not recognize motor difficulties as a significant element in ASD, despite the fact that the majority of cases (close to 90%) present difficulties, challenges, and/or abnormalities in their coordination, postural control, and motor performance. Moreover, the motor signs presented by infants, later diagnosed with ASD (nowadays), can be observed well before subsequent behavioral, communicative, or social symptoms can be detected. In fact, in their first-hand testimonials, individuals with ASD describe their physical experience in a way that explains the same phenomenon: difficulty in understanding and controlling their own body. As a result of misunderstood signals incoming from the body as well as those sent by their brain to their body, these individuals are unable to activate their body in a well-coordinated (understandable by an external observer) manner. In infants, this phenomenon prevents the formation of initial communication acts (reafference) in response to signals sent by an adult, which also impedes the development of accepted communicational signs and effective interactions with the communicating partner (the infant’s parents and caregivers). The inability to receive incoming messages and respond with outbound coherent messages prevent the developing infant from constructing meaningful and reliable communicational messages, thereby, prompting the development of compensations. At a later age, external observers interpret these flowed communicational\verbal and\or social difficulties, (which are based on sensory-motor difficulties) as symptoms of ASD.

Findings from interventional studies also show that motor interventions improve core ASD symptoms, in some cases with modest long-term effects [61,62], with seemingly similar results in young adults with ASD [63]. The accumulated evidence supports the hypothesis that sensory\proprioceptive-motor difficulties constitute the backbone of ASD and may be one of the key missing pieces of the puzzle known as ASD. Since motor and sensory elements influence one another, based on the findings presented here, early diagnosis (at 2–4 months of age), based mainly on motor signs, followed by early sensory–physical (therapy) interventions, can be effectively implemented at the early age of several months, with the potential to reduce future ASD symptoms and enhance the ability of individuals with ASD to integrate into society.

A word of caution: The perspective of the current author might be biased due to being a physical therapist focusing on the motor issues of individuals with ASD.

## Figures and Tables

**Table 1 biomedicines-12-01235-t001:** Symptoms observable in children with ASD that may be attributed to movement disorders.

#	Dysfunctions Typical of ASD and Attributed to Social/ Emotional/Behavioral Difficulties in DSM5	Motor Issues That Alternatively Explain the Observed Difficulty or Dysfunction	Source
1.	Delayed smile and amimia, odd facial expressions	Lack of control over facial musculature, difficulty coordinating facial muscles	[28]
2.	Not interacting in social settings	Difficulty coordinating head movement to follow a conversation	[29]
3.	Failure to cuddle	Muscle tone issues from childhood, proprioceptive issues	[30,31]
4.	Difficulty imitating others	Muscle tone issues from childhood; proprioceptive issues	[30,31]
5.	Lack of ability to initiate or an extreme delay in responding to others	Bradykinesia	[18,19]
6.	Stereotypical movements	Brain circuits, involved in motor stereotypes	[32]
7.	Odd hand and body postures	Disengagement between thought and motor function	[31,33]
8.	Anxiety	Misinterpretation of own body sensations	[34,35]
9.	Repetitive manipulation of objects	Provides a motor sense of security and control	[36]
10.	Verbal difficulties	Challenges in oro-motor verbal tasks	[37,38]
11.	Abnormal posture in sitting standing and walking	Asymmetry and irregularities in general movements at age 0–20 weeks	[8,9]
12.	Slowness of movement	Bradykinesia and muscle tone issues from childhood	[18,30]

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
