# Peer review of "Barking Up the Wrong Tree—Motor–Sensory Elements as Prodrome in Autism"

_biomedicines, 2024, doi:10.3390/biomedicines12061235_

Round 1
Reviewer 1 Report
Comments and Suggestions for Authors
Dear Author,
Thank you for submitting your manuscript ID 'biomedicines-2998393,' titled “Barking Up the Wrong Tree – Motor-Sensory Elements as Prodrome in Autism” for review. This opinion article is aptly suited for the 'Molecular and Translational Medicine' section of the special edition titled ‘Diagnostic Biomarkers and Novel Therapeutics Targets for Fragile X Syndrome, Autism Spectrum Disorders, and Genetic Neurodevelopmental Diseases: Advances and Challenges.’
Your paper presents a compelling hypothesis on early motor-sensory symptoms as indicators for Autism Spectrum Disorder (ASD). It is well-supported by a combination of recent studies and historical observations, with citations that are generally relevant and contemporary, enhancing the credibility of your claims.
However, the impact of your manuscript could be further enhanced by incorporating more direct evidence linking motor-sensory interventions to long-term outcomes in ASD. Additionally, the text would benefit from minor modifications to improve clarity and flow. I suggest varying sentence structures and reducing passive voice usage to make the content more engaging.
Specific areas requiring attention are outlined as follows:
1. Language and Presentation:
1.1. Simplify complex sentences and reduce the use of passive voice to make the article more accessible, particularly for readers who may not be specialists in the field.
1.2. Ensure consistency in the use of terms throughout the paper, particularly those describing ASD symptoms or interventions, to avoid potential confusion.
2. Detailing Selection Criteria:
2.1. Clarify the criteria used for selecting studies included in the opinion paper, including defining inclusion and exclusion criteria, to help readers understand the scope and relevance of the cited literature.
2.2. Consider incorporating different types of studies, such as case-control or cohort studies, to strengthen the findings. Discuss potential biases in the reviewed studies and their impact on the conclusions.
3. Comparative Analysis:
3.1. Discuss and compare alternative explanations for early symptoms of ASD to situate your hypothesis within the broader scientific debate.
3.2. Address potential counterarguments and provide a balanced view of why motor-sensory interventions could be more effective than other interventions.
4. Enhanced Discussion Section:
4.1. Expand on how these findings could alter current diagnostic or therapeutic practices in real-world settings, making the research more applicable to practitioners.
4.2. Clearly outline specific future research questions that arise from this study's findings to guide subsequent investigations.
5. Analytical Rigor (if applicable or possible):
5.1. Provide more detailed analyses of the reviewed studies, such as effect sizes or confidence intervals, to lend more weight to the conclusions.
6. References and Citations:
6.1. Ensure all references are up-to-date and relevant, removing any that are outdated or less relevant to the hypothesis.
I appreciate the effort you have put into this manuscript and look forward to its development. By addressing these points, your manuscript can achieve a higher degree of scientific rigor, broader relevance, and clearer presentation, thereby making it a more valuable contribution to the field of autism research.
Comments on the Quality of English LanguageThe English language used in the article is largely coherent, with a professional and academic tone appropriate for a Biomedicines journal.
There are minimal spelling and grammatical errors; however, the text could benefit from slight modifications for improved clarity and flow, such as varying sentence structures and reducing passive voice usage.
Author Response
Response to Reviewer 1
Journal: Biomedicines (ISSN 2227-9059)
Manuscript ID: biomedicines-2998393
Type: Opinion
Title: Barking Up the Wrong Tree – Motor-Sensory Elements as Prodrome in Autism
Author: Meir Lotan *
Section: Molecular and Translational Medicine
Special Issue:
Diagnostic Biomarkers and Novel Therapeutics Targets for Fragile X Syndrome, Autism Spectrum Disorders and Genetic Neurodevelopmental Diseases: Advances and Challenges
Submission Date: 20 April 2024
Date of this review: 03 May 2024 05:52:45
- Incorporate more direct evidence linking motor-sensory interventions to long-term outcomes in ASD.
Dear reviewer, thank you for this remark – a search was done in scientific resources. Very few articles examined long-term effects, yet as a result of these findings, the following 3 articles were found and incorporated within the text (at the conclusion part of the article), despite the fact that they measured long-term results, only 4 weeks post-intervention.
Ketcheson L, Hauck J, Ulrich D. The effects of an early motor skill intervention on motor skills, levels of physical activity, and socialization in young children with autism spectrum disorder: A pilot study. Autism. 2017 May;21(4):481-92. doi: 10.1177/1362361316650611. Epub 2016 Jun 26. PMID: 27354429.
Bremer E, Balogh R, Lloyd M. Effectiveness of a fundamental motor skill intervention for 4-year-old children with autism spectrum disorder: A pilot study. Autism 2014, 19, 980–91.
Shahane V, Kilyk A, Srinivasan SM. Effects of physical activity and exercise-based interventions in young adults with autism spectrum disorder: A systematic review. Autism. 2024 Feb;28(2):276-300. doi: 10.1177/13623613231169058. Epub 2023 May 1. PMID: 37128159.
- The text would benefit from minor modifications to improve clarity and flow. I suggest varying sentence structures and reducing passive voice usage to make the content more engaging.
Dear reviewer, thank you for this remark - Changes in language were performed throughout the text as suggested (see attached article with corrections)
- Simplify complex sentences and reduce the use of passive voice to make the article more accessible, particularly for readers who may not be specialists in the field.
Dear reviewer, thank you for this remark - this was done throughout the text (see attached corrected article)
- Ensure consistency in the use of terms throughout the paper, particularly those describing ASD symptoms or interventions, to avoid potential confusion.
Dear reviewer, thank you for this remark – The term ASD was used throughout the article.
- Clarify the criteria used for selecting studies included in the opinion paper, including defining inclusion and exclusion criteria, to help readers understand the scope and relevance of the cited literature.
Dear reviewer, thank you for this remark – As this is an opinion article, the articles were chosen according to their ability to lead the main reasoning of the article.
- Consider incorporating different types of studies, such as case-control or cohort studies, to strengthen the findings. Discuss potential biases in the reviewed studies and their impact on the conclusions.
Dear reviewer, thank you for this remark– I think that discussing the different articles, their strengths, and weaknesses will deflect the reader from the main thread of thought and prevent a smooth reading experience. However, I added a comment within the discussion section attesting to this point. “A word of caution: The perspective of the current author might be biased due to being a physical therapist focusing on the motor issues of individuals with ASD.
- Discuss and compare alternative explanations for early symptoms of ASD to situate your hypothesis within the broader scientific debate.
Dear reviewer, thank you for this remark– the whole article describes the current situation and why it needs a second observation. For instance, the opening section of the article (lines 20-41), and the whole middle section of the article, including the table (lines: 180-199), discusses the way ASD is looked upon today, with possible explanation for alternative way of looking at the current concepts and suggest a new way of thinking in regards to ASD diagnosis and treatment.
- Address potential counterarguments and provide a balanced view of why motor-sensory interventions could be more effective than other interventions.
Dear reviewer, thank you for this remark – the last part of the article (lines 235-255) suggests that motor intervention can significantly influence core elements of ASD. It is not possible to report the results of sensory-motor intervention in infancy to babies with ASD, as they are not diagnosed at a young age. The whole idea of this article is to change the way things are currently being performed. If the concept is changed and a shift from the current paradigm to a more motor-oriented approach is accepted, and if early diagnosis is performed, there and then the possibility of performing this type of motor intervention at an early age will become possible.
- Expand on how these findings could alter current diagnostic or therapeutic practices in real-world settings, making the research more applicable to practitioners.
Dear reviewer, thank you for this remark – As this is not a report regarding a specific research project, it is not yet applicable to practitioners. If this article is published, and the suggested approach is implemented, and early diagnosis is found possible, then intervention projects will be performed and should be written in a way that enables clinicians to actually apply what this article suggests.
- Clearly outline specific future research questions that arise from this study's findings to guide subsequent investigations.
Dear reviewer, thank you for this remark – A new section regarding futuristic steps regarding the new viewpoint of the current article was added (lines 256-281)
- Provide more detailed analyses of the reviewed studies, such as effect sizes or confidence intervals, to lend more weight to the conclusions.
Dear reviewer, thank you for this remark – this is a good suggestion, yet as scientific writing is concise and accurate, such an extension to the article will make it difficult for the reader to follow the new line of thought and will complicate the transfer of the point which this article is suggesting.
- Ensure all references are up-to-date and relevant, removing any that are outdated or less relevant to the hypothesis.
Dear reviewer, thank you for this remark – most of the cited articles are up-to-date expect for basic articles which are the hallmark of ASD such as the articles written by Kanner (1943), and Asperger (1991).
- There are minimal spelling and grammatical errors; however, the text could benefit from slight modifications for improved clarity and flow, such as varying sentence structures and reducing passive voice usage.
Dear reviewer, thank you for this remark – as can be seen un the text of the resubmitted article the whole article has been reevaluated for grammatical errors.
Reviewer 2 Report
Comments and Suggestions for Authors
Dear Author. Congratulations. The subject of this work is highly relevant, and I am recommending its acceptance. The interest in knowing more about autism has increased over the years. Please, see in the attached file my suggestions to try to improve the presentation of the work.
A concern, in general, I suggest avoiding the use of personal format (I, he, we, our, his) in the sentences throughout the manuscript.

It is fine.
Author Response
Response to Reviewer 1
Journal: Biomedicines (ISSN 2227-9059)
Manuscript ID: biomedicines-2998393
Type: Opinion
Title: Barking Up the Wrong Tree – Motor-Sensory Elements as Prodrome in Autism
Author: Meir Lotan *
Section: Molecular and Translational Medicine
Special Issue:
Diagnostic Biomarkers and Novel Therapeutics Targets for Fragile X Syndrome, Autism Spectrum Disorders and Genetic Neurodevelopmental Diseases: Advances and Challenges
Submission Date: 20 April 2024
Date of this review: 02 May 2024 19:30:30
- A concern, in general, I suggest avoiding the use of personal format (I, he, we, our, his) in the sentences throughout the manuscript. –
Dear reviewer, thank you for this comment – your suggestion has been implemented throughout the article (see attached corrected manuscript)
Reviewer 3 Report
Comments and Suggestions for Authors
Dear Author,
Thank you for the opportunity to read the manuscript, which is a single-author opinion on the analysis of the genesis of a wide spectrum of autism symptoms.This opinion article focuses on the sensory-motor issues of autistic individuals and suggests a new diagnostic direction that considers sensory-motor elements as the core of ASD. It is widely known that early treatment is known to improve the prognosis of autistic children, therefore early diagnosis is crucial - which underlines the importance of the opinion presented. If the author is correct, perhaps early motor problems may also be at the root of ASD. In the light of the increasing number of people with ASD, and the relatively late age of its full diagnosis, the early motor elements of autism may provide very early (if not the first) valuable diagnostic information.
In my opinion, this article can be very useful for broadly understood public health. It also significantly strengthens the opinions of other specialists in the field in terms of the importance of motor difficulties as the basis of later appearance of social and communicative difficulties.
Author Response
Response for reviewer 3:
Journal: Biomedicines (ISSN 2227-9059)
Manuscript ID: biomedicines-2998393
Type: Opinion
Title: Barking Up the Wrong Tree – Motor-Sensory Elements as Prodrome in Autism
Author: Meir Lotan *
Section: Molecular and Translational Medicine
Special Issue:
Diagnostic Biomarkers and Novel Therapeutics Targets for Fragile X Syndrome, Autism Spectrum Disorders and Genetic Neurodevelopmental Diseases: Advances and Challenges
Submission Date: 20 April 2024
Date of this review: 02 May 2024 20:09:27
Thank you for reviewing my article.
Professor Meir Lotan
Bottom of Form
Round 2
Reviewer 1 Report
Comments and Suggestions for Authors
Dear Author,
I would like to express my sincere gratitude for the privilege of reviewing the second version of your manuscript. After reviewing the response letter, I agree with the improvements made in this version. The submitted manuscript is a valuable addition to the special issue entitled "Diagnostic Biomarkers and Novel Therapeutic Targets for Fragile X Syndrome, Autism Spectrum Disorders, and Genetic Neurodevelopmental Diseases: Advances and Challenges."
You have made commendable efforts to improve the quality of the manuscript. I have no additional questions and appreciate the opportunity to contribute to the review process. Thank you again for your commitment to advancing scientific knowledge in the field of autism.